# Novel Therapies to Address Unmet Needs in ITP

**DOI:** 10.3390/ph15070779

**Published:** 2022-06-23

**Authors:** María Eva Mingot-Castellano, José María Bastida, Gonzalo Caballero-Navarro, Laura Entrena Ureña, Tomás José González-López, José Ramón González-Porras, Nora Butta, Mariana Canaro, Reyes Jiménez-Bárcenas, María del Carmen Gómez del Castillo Solano, Blanca Sánchez-González, Cristina Pascual-Izquierdo

**Affiliations:** 1Hematology Deparment, Hospital Universitario Virgen del Rocío, Instituto de Biomedicina de Sevilla (IBIS/CSIC), 41013 Sevilla, Spain; 2Hematology Deparment, Hospital Universitario de Salamanca, 37007 Salamanca, Spain; jmbastida@saludcastillayleon.es (J.M.B.); jrgp@usal.es (J.R.G.-P.); 3Hematology Deparment, Hospital Universitario Miguel Servet, 50009 Zaragoza, Spain; gcaballeron@salud.aragon.es; 4Hematology Deparment, Hospital Universitario Virgen de las Nieves, 18014 Granada, Spain; laura_eu@hotmail.com; 5Hematology Deparment, Hospital Universitario de Burgos, 09006 Burgos, Spain; tjgonzalez@saludcastillayleon.es; 6Hematology Deparment, Hospital Universitario La Paz-Idi-PAZ, 28046 Madrid, Spain; nbutta@hotmail.com; 7Hematology Deparment, Hospital Son Espases, 07210 Palma, Spain; mcanaro@gmail.com; 8Hematology Deparment, Hospital Serranía de Ronda, 29400 Ronda, Spain; rejimbar@hotmail.com; 9Hematology Deparment, Complexo Hospitalario Universitario A Coruña, 15006 A Coruña, Spain; ma.del.carmen.gomez.del.castillo.solano@sergas.es; 10Hematology Deparment, Hospital del Mar, 08003 Barcelona, Spain; bsanchezgonzalez@hospitaldelmar.cat; 11Department of Hematology, Hospital Gregorio Marñón (HGUGM), Instituto de Investigación Gregorio Marañón, 28006 Madrid, Spain; crisizquierdo3@yahoo.es

**Keywords:** immune thrombocytopenia, thrombopoietin, autoantibodies, platelets, targeted therapies

## Abstract

Primary immune thrombocytopenia (ITP) is an autoimmune disorder that causes low platelet counts and subsequent bleeding risk. Although current corticosteroid-based ITP therapies are able to improve platelet counts, up to 70% of subjects with an ITP diagnosis do not achieve a sustained clinical response in the absence of treatment, thus requiring a second-line therapy option as well as additional care to prevent bleeding. Less than 40% of patients treated with thrombopoietin analogs, 60% of those treated with splenectomy, and 20% or fewer of those treated with rituximab or fostamatinib reach sustained remission in the absence of treatment. Therefore, optimizing therapeutic options for ITP management is mandatory. The pathophysiology of ITP is complex and involves several mechanisms that are apparently unrelated. These include the clearance of autoantibody-coated platelets by splenic macrophages or by the complement system, hepatic desialylated platelet destruction, and the inhibition of platelet production from megakaryocytes. The number of pathways involved may challenge treatment, but, at the same time, offer the possibility of unveiling a variety of new targets as the knowledge of the involved mechanisms progresses. The aim of this work, after revising the limitations of the current treatments, is to perform a thorough review of the mechanisms of action, pharmacokinetics/pharmacodynamics, efficacy, safety, and development stage of the novel ITP therapies under investigation. Hopefully, several of the options included herein may allow us to personalize ITP management according to the needs of each patient in the near future.

## 1. Introduction

Primary immune thrombocytopenia (ITP) is an autoimmune disease that leads to a decreased platelet count with the resulting hemorrhagic risk. The overall incidence is about 3.3 per 100,000 adults/year [1]. Although the pathways leading to platelet destruction have been identified, the mechanism underlying imbalances of immune system regulation is not well understood. As in other autoimmune diseases, an increased T helper (Th) cell type 1 (Th1)/Th2 ratio can be observed. There is evidence suggesting that this change is preceded by an anomalous increase in Th17 cells and suppression of the activity of regulatory T cells (Treg) [2]. The immune dysregulation triggers a variety of actions, resulting in accelerated platelet clearance and reduced platelet formation (Figure 1). B cells produce autoantibodies against platelet glycoproteins and other platelet structures. Opsonized platelets are phagocyted by macrophages, especially in the spleen. Phagocytes also act as antigen-presenting cells (APCs). At this point, the Th cells further stimulate the B cells to produce more autoantibodies. Conversely, opsonized platelets are no longer protected against the neuraminidase-dependent desialylation of the O-glycans of surface glycoproteins [3]. The desialylated platelets interact with hepatocytes via Ashwell–Morell receptors and are finally destroyed by Kupffer cells [4]. Opsonized platelets also activate the complement system, while the immune imbalance increases the activity of cytotoxic T cells, both actions contributing further to reduced platelet counts. Finally, autoantibodies also interfere with megakaryocyte maturation and reduce thrombopoietin levels, thus challenging platelet production in the bone marrow [5].

Bleeding generally comprises hematoma, purpura, petechiae, mouth and nose hemorrhages, and heavy menstrual bleeding or urogenital bleeding, and it is usually of mild to moderate severity. Nevertheless, intracerebral hemorrhage (ICH) and other forms of severe bleeding have been observed to occur in up to 1% and 15% of ITP cases [6]. Although there is no consensus about autoantibody assessment during the diagnostic procedure, an early determination is advisable since a positive result may guide further steps and allow prompt therapy initiation [7]. The current first-, second- and third-line treatments are not effective in a non-negligible proportion of patients, especially in the long term. After revising these therapies and identifying their main limitations, this review will focus on a series of emerging therapies that may improve the outcome and quality of life of ITP patients in the near future.

## 2. Currently Approved Treatments to Manage ITP

### 2.1. First-Line Treatments

The goal of ITP treatment is to reduce the bleeding risk by raising platelet counts and induce remission by restoring the immune balance. Treatment should be individualized according to the individual patient’s risks, age, and the toxicity associated with therapeutic options. ITP treatment is intended to increase platelet count to prevent bleeding, induce remission, and improve the patient’s quality of life. First-line treatment consists of the administration of corticosteroids, in order to decrease platelet destruction by avoiding autoantibody generation and excess cytotoxic T cell activity. Rescue treatments may be also used, with nonspecific intravenous immunoglobulins (IVIG) in cases of severe bleeding, and platelet transfusion or thrombopoietin receptor agonists (TPO-RAs) in cases of life-threatening bleeding (Figure 2). The treatment of elderly patients may be challenging. On the one hand, they are at increased risk of bleeding. On the other hand, they are often at risk of venous or arterial thrombosis, thus requiring concomitant anticoagulant or antiplatelet therapy. Finally, the use of immunosuppressants exposes patients to a substantial risk of infection [8]. 

As seen in Appendix A [9], corticosteroids are generally well tolerated, although weight gain, hypertension, or hyperglycemia, among others, are possible adverse effects of these drugs, especially after prolonged periods of treatment. Information regarding their pharmacokinetic and pharmacodynamic details, as well as the recommended dosage, response and side effects, is given in Table 1, Table 2 and Table 3. The main limitation of corticosteroids is that, although they may be effective in the short term, the response is poorly sustained after drug discontinuation. Overall, up to 70% of patients experience relapse in the long term, thus requiring additional treatment to prevent bleeding [8,10,11]. Finally, likelihood of *Helicobacter pylori* infection should be evaluated in the appropriate geographical areas and treated in the case of a positive test because of the possibility of ITP occurrence secondary to this infection and the low incidence of adverse events related to this recommendation [10].

### 2.2. Second-Line Treatments

According to the current guidelines, the treatments that are considered second-line choices include the thrombopoietin receptor agonists (TPO-RAs), rituximab, fostamatinib and a splenectomy [8,10,11]. The choice of treatments is conditioned by the patients’ risk of comorbidities, mainly infection or thrombosis, and their preferences (avoidance of surgery or long-term treatment) [5,8,10,11].

#### 2.2.1. TPO-RAs

TPO-RAs mimic the action of endogenous TPO on megakaryocytes and megakaryocyte precursors, promoting their survival, growth, and maturation. Therefore, TPO-RAs augment platelet production to compensate for the increased platelet turnover (Appendix A). TPO-RAs enter the bone marrow and bind the TPO receptor (TPO-R) stimulating megakaryopoiesis and platelet production [12,13]. Endogenous TPO induces conformational changes in TPO-R that lead to the downstream activation of the JAK-STAT signaling pathway and the phosphorylation of phosphatidylinositol 3-kinase (PI3K) and MAP kinases (MAPK) [14,15]. The subsequent mobilization of hematopoietic stem cells, immature megakaryocytes and erythroid precursors leads to mature megakaryocyte formation. TPO-RAs mimic the physiological actions that human TPO would perform in a healthy person [15,16]. Currently, four of them are approved worldwide: romiplostim, eltrombopag, avatrombopag and lusutrombopag, although the latter is indicated only for patients with chronic liver disease who are undergoing invasive procedures [17,18,19]. There is a fifth TPO-RA under development in China, the so-called hetrombopag olamine, which will be addressed in the next section. Romiplostim is a peptidic TPO mimetic for subcutaneous administration, which shows a plasma half-life that is longer than that of the other TPO-RAs, while eltrombopag and avatrombopag are non-peptidic TPO mimetics, and are taken orally. Romiplostim binds the extramembrane domain of the TPO-R, the so-called MPL, in a similar way to endogenous TPO, thus activating signal transduction via STAT, PI3K/AKT, and MAPK. However, eltrombopag and avatrombopag interact with the transmembrane domain of MPL, initiating a signal transduction that is only dependent on STAT phosphorylation [20,21]. These two different mechanisms of action may explain the lack of cross-resistance after the sequential use of TPO-RAs. Table 1, Table 2 and Table 3 provide details about the pharmacokinetics (PK), pharmacodynamics (PD), dosage, efficiency and safety of these drugs. Further information is collected in Appendix A. Regardless of the TPO-RA chosen, these agents have an overall response rate of 70–80%, which is, thus, higher than that of other treatments, although more prolonged periods of treatment are typically required to achieve this goal. Patients presenting with a relapse after a splenectomy have lower response rates of between 40 and 60% [22,23,24]. TPO-RAs do not need to be continued indefinitely; about 33% and 8–32% of patients have sustained remission after one or two years, respectively [25,26,27,28,29], thus not requiring the prolongation of treatment.

TPO-RAs are generally well-tolerated in ITP patients (Table 3) [22,30,31]. Transaminase monitoring is advised for patients on eltrombopag, due to the risk of hepatotoxicity [32]. Neither platelet hyperreactivity nor spontaneous platelet aggregation has been observed [33,34], and there does not seem to be a relevant risk of venous or arterial thrombosis [22,23,35,36]. There is no risk of irreversible marrow collagen fibrosis in patients receiving TPO-Ras for extended periods of time [27,37].

TPO-Ras have been proposed as a first-line choice, although the latest guidelines do not recommend this practice yet. A study showed that romiplostim enabled the avoidance of relapse in 30% of patients after treatment discontinuation [25]. On the other hand, promising preliminary data have also been obtained by combining eltrombopag with dexamethasone [38].

#### 2.2.2. Rituximab

Rituximab is a monoclonal antibody directed against a glycoprotein expressed on the surface of mature B cells, namely, CD20. The promise of rituximab to manage autoimmune conditions relies on its ability to induce a fast and deep, but reversible, depletion of B cells [39]. Numerous studies have evaluated the usefulness of rituximab to treat ITP. An updated review of clinical trials and real-world practice regarding this topic was performed by Lucchini et al. Unfortunately, although initial response rates are good in up to 60% of patients, only 20–30% of these patients exhibit a sustained remission in the long term after discontinuation, irrespective of the timing and/or dose schedules used [40]. Rituximab failure may be partly explained by the emergence of long-lived splenic plasma cells, due to the presence of cytokines such as the B cell-activating factor (BAFF) [41], as well as the important role played by CTL in the destruction of megakaryocytes and platelets [42]. Another disadvantage of rituximab is that there are no biomarkers predictive of response. On the other hand, rituximab may be especially useful at the early stages of the disease, particularly in young women. Its use in combination with other drugs may provide additional positive effects as well [40]. The safety profile of rituximab use in ITP patients has been satisfactory until now. Table 1, Table 2 and Table 3 provide detailed information about the main characteristics of rituximab in the context of ITP management.

#### 2.2.3. Fostamatinib

Autoantibody-coated platelet phagocytosis by the reticuloendothelial system (RES) is one of the most important mechanisms involved in ITP pathogenesis; therefore, it provides attractive therapeutic targets. When the opsonizing immunoglobulin binds to the Fc receptors (FcR) of macrophages, cytoskeleton rearrangements allow these to initiate phagocytosis. Cell changes are mediated by a signaling cascade involving several kinases. Among these, spleen tyrosine kinase (Syk) plays a central role by activating phagocytosis mediators, such as the immunoreceptor-activating motifs (ITAMs). SYK is a non-receptor tyrosine phosphorylating enzyme that is responsible for triggering signaling pathways via ERK, p38 or JNK. Fostamatinib is an orally administered prodrug that was designed to deliver the active metabolite R940406 (R406), which is a Syk inhibitor (Table 1, Table 2 and Table 3, Appendix A). Fostamatinib has been recently approved for the treatment of chronic ITP. Its authorization is based on the results of two phase-III trials, FIT-1 and FIT-2, in adult patients with persistent or chronic primary ITP [43]. Patients with an insufficient response to at least one prior treatment with corticosteroids, TPO-RA, splenectomy, or/and IVIG, and with platelet counts < 30 × 10^9^/L, were recruited to receive 100 mg of fostamatinib orally twice daily, which could be increased to 150 mg if no response was observed after 4 weeks. Forty-six percent of patients received concomitant medication, mainly corticosteroids. Eighty-six percent of patients required a higher fostamatinib dose. Forty-three percent of patients had at least one platelet count ≥ 50 × 10^9^/L in the first 12 weeks of treatment (vs. 14% among those on the placebo). Platelet response, established in counts ≥ 50 × 10^9^/L on ≥4 of the last 6 visits between weeks 14 and 24, was achieved by 16.8% of the patients (vs. 2.1% in those receiving the placebo). In a long-term extension study, with a mean duration of 19 months, 75% of patients discontinued treatment due to a lack of response or adverse effects in 34% or 21% of cases, respectively [44]. Patients were most likely to respond when fostamatinib was used as a second- rather than as a third (or more)-line therapy [45]. Regarding the safety profile, although most adverse events were mild or moderate, the discontinuation rate due to these events was 21% in the aforementioned studies. Furthermore, it must be noted that fostamatinib has a detrimental effect on osteogenesis in patients under 18 years of age who have not reached full skeletal maturity, as well as on ossification in any patient after a fracture. Finally, its antiangiogenic action precludes its use during pregnancy.

#### 2.2.4. Splenectomy

Until recently, splenectomy was considered to be the best option for those patients who were unresponsive to corticosteroids, with a sustained response rate ranging between 60 and 70% over 5–10 years, which is higher than that of other therapies [46]. Nevertheless, the development of drugs such as TPO-RAs and others, the use of which could avoid patient exposure to surgical risks and complications, namely, bleeding, infection, sepsis and venous or arterial thromboembolic events, means that its use is limited to those chronic ITP patients who are refractory to, or who are unable to use, the available pharmacological options.

Splenectomy should be avoided during the first 12 months after diagnosis because, during that time, spontaneous remission or disease stabilization may occur. Indeed, the good results of splenectomy are not unexpected, since the spleen is the main site for autoantibody generation and platelet destruction (Appendix A). However, the identification of those ITP patients who would benefit most from this procedure has been precluded by the lack of reliable predictors of response. Platelet scintigraphy studies have been described as a useful tool for the purpose in some groups, with a high response rate in patients with a splenic profile that is close to pure. Despite this, the results of splenectomy are controversial and there is low availability, so the practice is not recommended in the guidelines [46]. Nevertheless, a recent study showed that an age under 41 years, a body mass index (BMI) < 24.3 kg/m^2^ and platelet counts ≥ 97 × 10^3^/µL are independent remission prognostic factors [47]. Finally, an increased relapse rate, together with the requisite exposure to serious surgical complications, make splenectomy an inadvisable practice in the elderly [7].

### 2.3. Third-Line Treatments

Immunomodulatory agents, such as mycophenolate mofetil, cyclosporin A or hydroxychloroquine, other steroids such as danazol, nucleic acid synthesis inhibitors such as azathioprine, and antibacterial agents such as dapsone or cell growth inhibitors such as vincristine, vinblastine or cyclophosphamide may be considered if the first- and second-line choices were not effective or induced non-acceptable adverse events. Nevertheless, their low rates of complete response, as well as the risk of severe side effects, preclude their widespread use [17,18]. Details about the management of some of them are shown in Appendix A.

## 3. Novel Drugs and Therapies to Treat ITP

The aforementioned rates of sustained response obtained with the currently approved therapies suggest that the optimization of ITP patient management is necessary (Table 4) [48]. This section focuses on those ITP treatments that are currently under investigation or that are close to regulatory approval, paying attention to how their mechanisms of action or basic PK/PD characteristics may contribute to filling the gaps in ITP management. When available, preliminary trial results are also discussed. A first summary of the new approaches to ITP treatment currently under clinical investigation, including trial references, is shown in Table 5.

### 3.1. Other TPO-RAs: Hetrombopag Olamine

Hetrombopag olamine (hetrombopag) is a small molecule, non-peptide TPO receptor agonist, which is in the same class as eltrombopag. Available data regarding its PK and PD features are shown in Table 6 and Table 7 [49,50,51]. The first in vitro studies showed that it was able to induce the proliferation of a TPO-dependent murine cell line through the activation of the STAT, PI3 kinase and ERK pathways while preventing apoptosis by restoring the Bcl-xL/BAX ratio [52]. Accordingly, hetrombopag enhanced the proliferation of human CD34+ cells into megakaryocytes and stimulated proplatelet production, its effect being additive to that of TPO. In vivo, it must be borne in mind that the absorption and biological activity of hetrombopag is reduced by fatty calorific food [51]. Like other TPO-RAs, hetrombopag is administered orally and daily. A phase-3 trial and the corresponding extension study showed good platelet response rates of about 85% (vs. 22% when patients were administered placebo), which were almost the same (84 and 86%) with the 2.5 and 5 mg doses, respectively. Relevant adverse events were reported in 18% of patients [53]. Hetrombopag does not influence platelet aggregation and is considered to be tolerated well, with a manageable safety profile. Interestingly, toxicological experiments have shown that the toxicity of hetrombopag is lower than that of eltrombopag, its hepatotoxicity being undetectable [50]. Nevertheless, the assessment of its efficacy in the long term, which is mandatory for establishing whether or not its use should be prioritized over the TPO-RAs that are currently used, is still ongoing. Currently, approval to use hetrombopag as a second-line treatment for ITP is limited to China.

### 3.2. Inhibitors of Kinases Involved in Phagocyte Cytoskeleton Rearrangements

#### 3.2.1. Syk Inhibitors

Although the only currently approved drug in this class is fostamatinib, there are two highly selective Syk inhibitors under development, which are aimed at treating relapsed/refractory ITP patients.

##### HMPL-523

This is an orally available small molecule that potently inhibits Syk reversibly (Appendix A). Preliminary PK studies showed that HMPL-523 was rapidly absorbed under both fasted and fed conditions. In fact, food consumption seemed to increase its bioavailability. The time to reach the peak plasma concentration (Tmax) was slightly longer than that for fostamatinib (Table 6 and Table 7). The same PK/PD analyses concluded that exposing ITP patients to daily ≥ 200 mg doses should provide the target coverage required for clinical efficacy [54]. A phase-1B trial (NCT03951623) where 33 patients received daily doses of 100–400 mg over 24 weeks showed that HMPL-523 was tolerated well (although there were increases in aspartate aminotransferase (AST), alanine aminotransferase (ALT) and lactate dehydrogenase (LDH) in 20–25% of participants). The overall response ranged between 30 and 70% and was not dose-dependent: 50, 33.3, 68.8 and 33.3% at doses of 100, 200, 300 and 400 mg, respectively. Thirty-one percent of those patients who had received a dose of 300 mg showed a durable response (vs. 9.1% in the placebo group in both analyses) [55]. A phase-3 trial (NCT05029635) is currently ongoing. Provided that the results of this trial are encouraging, a comparison with fostamatinib would be of interest to establish if HMPL-523 could be the Syk inhibitor of choice.

##### SKI-O-703 (Cevidoplenib)

This is a low-molecular-weight synthetic molecule that is orally available and is able to inhibit SYK reversibly; it is currently undergoing phase 1 and 2 clinical trials (Appendix A). Its safety, PK and PD are being evaluated in a single ascending dose (NCT02717988) and multiple ascending doses (NCT03315494) in clinical studies in healthy subjects. It is also being evaluated in relapsed/refractory ITP patients (NCT04056195). The results are not yet available.

#### 3.2.2. Bruton’s Tyrosine Kinase Inhibitors

Bruton’s tyrosine kinases (BTK) are cytoplasmic proteins belonging to the family of tyrosine kinases expressed in hepatocellular carcinoma (TEC) kinases. BTKs are immediately downstream from Syk and play important signaling functions in B cells and myeloid cells. Once activated, their kinase activity enables signaling via multiple mediators, such as JAK2, STAT3, IkB or PLCγ2, and also as a part of the inflammasome. In the context of autoimmune disorders, BTKs contribute significantly to the release of inflammatory mediators and the enhancement of phagocytosis ability and antigen presentation in myeloid cells, as well as contributing to proliferation and plasma cell differentiation in autoreactive B cells (revised in [56]). Therefore, BTKs present attractive targets to reduce the platelet destruction in ITP.

##### Rilzabrutinib

Rilzabrutinib was the first small molecule for oral administration that was able to react covalently with BTK and inhibit it reversibly. These properties should allow rilzabrutinib to target the immune-mediated pathways affected in ITP (Appendix A). It must be remarked that, unlike other BTK inhibitors such as ibrutinib, rilzabrutinib does not interfere with normal platelet aggregation. Data regarding some PK and PD features of rilzabrutinib are available (Table 6 and Table 7) [57]. The results of a dose-finding phase-1 study that recruited 60 ITP patients who had relapsed or were refractory to at least one prior line of therapy (NCT03395210) have recently been reported. Rilzabrutinib showed a good safety profile at doses ranging between 200 mg once daily and 400 mg twice daily, with all treatment-related adverse events being transient and of grade ≤ 2. Interestingly, rilzabrutinib demonstrated rapid and sustained clinical activity that improved with the length of treatment. At a median of 167 days of therapy, 40% of patients demonstrated a platelet response. The median time to achieve a platelet count of ≥50 × 10^3^/µL was 11.5 days. The authors identified the dose of 400 mg twice daily as being most suitable for further trials [58,59]. Likewise, the first randomized, multicenter, phase-3 study with an open-label extension phase (LUNA3, NCT04562766) is underway, with the aim of evaluating the efficacy and safety of rilzabrutinib relative to placebo in adult and adolescent patients with persistent or chronic ITP [60]. The results of this trial may shed light on whether BTK inhibitors will be able to provide ITP patients with more durable protection than that achieved by the use of current or other emergent therapies.

##### Orelabrutinib

This is another potent, orally administered and highly selective BTK inhibitor. Orelabrutinib is a small molecule designed as a second-generation inhibitor to have improved BTK specificity. Unlike rilzabrutinib, orelabrutinib induces irreversible BTK inhibition, with a half-maximal inhibitory concentration (IC50) of 1.6 nM. Preliminary PK and PD data are encouraging (Table 7) and suggest that orelabrutinib target selectivity is superior to that of other BTK inhibitors. The mean terminal plasma half-life (t1/2) is ~4 h across the 20–400 mg dose range, with no accumulation in plasma after repeated dosing. Food interactions have not been reported either [61]. Once the results of the currently ongoing trials are available [62], a comparison with those obtained by reversible BTK inhibitors will allow researchers to assess whether or not second-generation irreversible inhibitors are superior for targeting BTK in ITP.

### 3.3. Inhibitors of Neonatal Fc Receptors

Neonatal Fc receptor (FcRn) is a beta-2-microglobulin-associated protein that is structurally related to the major histocompatibility class I (MHC-I) family members. FcRn plays an important role in preventing IgG degradation during transcytosis of circulating IgGs. In this process, FcRn-bound IgGs are released again to the extracellular fluid, while free IgG are cleaved in the lysosomal compartment (Appendix A) [63]. Indeed, the inhibition of FcRn should decrease IgG recycling, which means that, in the context of ITP, the number of circulating autoantibodies would decrease accordingly. Thus, an a priori therapy based on FcRn inhibition would reduce the extent of autoantibody-mediated actions (Appendix A).

#### Efgartigimod

Efgartigimod is a hIgG1-derived Fc fragment. It works as an FcRn blocker by intravenous administration and is one of the most widely investigated drugs within this class. The available PK and PD data are collected in Table 6 and Table 7. A phase-2 study, where ITP patients who were resistant to previous therapies received four weekly 5 or 10 mg/kg intravenous infusions, showed that efgartigimod was well tolerated while being able to rapidly reduce total IgG levels by about 64% with respect to baseline levels. This decrease was associated with clinically relevant increases in platelet counts: forty-six percent of patients on efgartigimod demonstrated a platelet count of ≥50 × 10^9^ /L on at least two occasions, and 38% achieved counts of ≥50 × 10^9^ /L for at least 10 cumulative days (vs. 25% and 0%, respectively, in the placebo group). The proportion of patients suffering from bleeding episodes was also reduced with efgartigimod [64]. The ongoing phase-3 trials should shed more light on the usefulness of this drug to manage ITP.

#### Rozanolixizumab

Rozanolixizumab is a humanized monoclonal antibody directed against FcRn, which is currently being studied in phase-3 trials. PK and PD data in humans have been made available (Table 6 and Table 7). In an animal model, rozanolixizumab was able to induce a dose-dependent reduction of up to 85% in IgG levels, without causing changes to the lymphocyte populations, cytokines or infection rate. A phase-1 randomized study in humans, where doses of 1.4 or 7 mg/kg were administered subcutaneously or intravenously, showed that the treatment induced a reduction in IgG levels of up to 50%, the higher effects being observed at 7 to 10 days [65]. The safety profile was favorable. No increase was detected in the infection rate, the main adverse event being headaches. Two phase-3 trials with rozanolixizumab in ITP patients are currently ongoing.

#### Other FcRn blockers

Another human anti-FcRN monoclonal antibody, the so-called HBM9161 (batoclimab), is currently being investigated in a phase 2/3 trial (NCT04428255). Likewise, chimeric proteins formed by the union of human IgG2 sequences and murine IgG2-Fc, with the ability to block the FcRn pathway, are currently being developed. Several of these molecules, the so-called Stradomer, HexaGard and SIF3, have been able to prevent or decrease immune platelet destruction in murine ITP models [48,66]. Antibodies against FcγRIIIA and the Fcγ soluble receptor (sFcγR) have also been useful in ITP rodent models [67].

### 3.4. Staphylococcal Protein A

Staphylococcal protein A (SpA) is a 42 kDa protein that can be found either in the bacteria cell wall or secreted. It binds to the Fab and Fc regions of IgG, reducing B-cell signaling and macrophage activation in animal models and promoting B-cell apoptosis [68,69].

#### PRTX-100

PRTX-100 is a highly purified form of Staphylococcal protein A that binds to FcγRIII (CD16) on human monocytes, thus preventing the phagocytosis of opsonized platelets (Appendix A). In a murine model of chronic ITP, its administration induced a dose-dependent rise in platelet count, comparable to that achieved with IVIG [70]. A small dose-escalation clinical trial was conducted with 6 adult patients, who were intravenously administered 4 weekly doses of 1 to 3 µg/kg. The safety profile was acceptable, although a platelet response was observed in only one patient who had been treated with the lowest dose [71,72]. Other trial results have not been reported. PK and PD information has not been made available yet.

### 3.5. Immunoglobulin-Based Therapies

#### 3.5.1. Hyper-Sialylated Immunoglobulin G

Although most Fc receptors activate phagocytosis, FcγRIIb exerts the opposite effect. IVIGs promote the upregulation of the FcγRIIb, which, in turn, prevents FcγRIII activation upon the binding ligand [73]. Scientific evidence has demonstrated the importance of glycosylation in immunoglobulin function, especially in the anti-inflammatory activity of IVIGs, which is dependent on the extent of Fc-sialylation. Therefore, it is conceivable that hyper-sialylated immunoglobulins could be used in ITP to prevent platelet phagocytosis more efficiently [74,75].

##### M254

M254 is a hyper-sialylated IgG, the preclinical data of which suggest that its activity may be 10-fold that of immunoglobulins, with unaltered glycosylation patterns. Preliminary studies with healthy volunteers did not find any relevant safety concerns [76]. An additional advantage of M254 may be that its high activity could allow minimizing the volume to infuse. A phase 1/2 clinical trial is currently ongoing. PK and PD reports have not been made available yet (Appendix A).

#### 3.5.2. Other Immunoglobulin-Derived Drugs

These therapies are in the preliminary stages. It is worth mentioning that three decades ago, there was a pilot study wherein Fc gamma fragments were infused into children with ITP, with the aim of blocking the Fcγ receptors. There was a response in 90% of the children. Accordingly, a rise in soluble FcγRIII was observed, which was proportional to the platelet count increase. Half of the patients maintained the response for more than a month [76]. Immunoglobulin-based molecules currently in ongoing phase-1 trials are PF06755347 (formerly GL-2045), which is a recombinant human IgG1-based Fc multimer designed to recapitulate the anti-inflammatory activities of IVIGs on the innate and adaptive immune responses [77,78], and M230 (CSL 730), which is a trimeric construct evolved from studies focused on understanding the relationship between IgG-Fc valency and binding to FcγR. Since such binding promotes the phosphorylation of many of the kinases involved in signaling, a number of recombinant Fc structures were designed and tested for their ability to activate or inhibit FcγR [79].

### 3.6. Complement Inhibitors

Inhibition of the classical complement pathway could be a new target in the treatment of ITP. The binding of autoantibodies to platelets activates the complement via the C1 complement receptor, which is expressed by macrophages. Although the importance of the role played by the overactivation of the classical complement pathway in ITP is not fully understood, it is known that deposits of the complement are present on the platelet surfaces in more than half of the patients with ITP [80] and that the complement activation correlates with disease activity [81]. The first in vitro experiments with TNT003, a monoclonal antibody that is able to block the C1s component of the classical complement pathway, showed that its use inhibited complement activation in the samples of ITP patients, reducing Cd, C3b and C5b-9 deposits. The findings of this study also supported a relevant role for the classical pathway in complement activation in ITP [82].

#### Sutimlimab

Two patients with severe refractory ITP showed a rapid increase in platelets after treatment with a plasma-derived human C1 esterase inhibitor [83], which encouraged researchers to explore the development of drugs with similar properties. Sutimlimab (BIVV009) is a humanized monoclonal antibody that is able to inhibit the C1s component of the complement (Table 6 and Table 7) (Appendix A). In vitro, sutimlimab selectively inhibits the activation of the classical pathway by binding to C1 and decreasing C3b deposition and the subsequent membrane attack complex formation. Furthermore, the decreased C3b deposition prompts autoreactive B cells to interact with the autoantigen via the B cell receptor (BCR), exclusively. As a result, B cell activation is limited. Two trials showed the efficacy of sutimlimab in cold agglutinin disease [84]. In the context of ITP, sutimlimab achieved a rapid and durable response in severe chronic ITP patients who were unresponsive to at least 2 prior therapies, in a phase-I trial (NCT03275454). Twelve patients received sutimlimab intravenously, weekly for the first two weeks and then every two weeks, until 21 weeks. After a 9-week washout period, the responders would continue with a long-term extension treatment. A response (platelets ≥ 50 × 10^9^/L) was identified fairly quickly in 42% of participants (2 days median value), and complete remission (platelets ≥ 100 × 10^9^/L, on 2 consecutive occasions at intervals longer than 7 days) was observed in 33% of cases. In the washout period, the platelets returned to baseline levels. Interestingly, the 6 patients who were included in the extension phase not only achieved complete remission but also maintained this for more than 1 year of therapy, while showing an acceptable safety profile, with no deaths, thromboembolic events or discontinuations [85]. Unfortunately, sutimlimab was unable to induce a response in ITP patients in a phase-2 trial (NCT04669600), which, in fact, was finally discontinued. From the current evidence, it can be concluded that the inhibition of the complement, albeit still offering a promising approach to managing ITP, may not work in all patients. In this scenario, the identification of biomarkers that are able to predict patient response reliably would be highly desirable.

### 3.7. Platelet Desialylation Inhibitors

The sialic acids on complex glycan molecules prolong the life span of proteins and cells. Young platelet surfaces are rich in sialic acid. The platelet membrane glycoproteins GPIIb/IIIa and GPIb/IX are rich in carbohydrate contents; GPIba is particularly sialylated, which means that it is profoundly modified when the desialylating activity increases [86]. Importantly, opsonized platelets are not protected against the desialylation of surface glycoproteins. Sialic acid is removed from platelets by sialidases, such as neuraminidase 1 and neuraminidase 3 [87], after which the platelets are cleared by the liver [88] via a mechanism led initially by hepatocytes, which recruit platelets through their Ashwell–Morell receptors, and finally by Kupffer cells via the C-type lectin receptor, CLEC4F. Recent studies have explored whether platelet desialylation might play a role in ITP. In mice, the administration of an antiplatelet antibody specific for GPIba but not for GPIIb/IIIa caused platelet activation, the externalization of the platelet neuraminidase and platelet desialylation [89]. CD8+ T cells also induce the desialyation of platelets in murine models [90]. Conceivably, the inhibition of neuraminidase-dependent platelet desialylation may prevent platelet destruction in the liver (Appendix A).

#### Oseltamivir

Oseltamivir is an orally administered small molecule with neuraminidase inhibitor activity, which has been approved for the treatment of influenza. The PK and PD features of oseltamivir have been reported (Table 6 and Table 7). Oseltamivir is able to improve platelet counts in mice treated with anti-GPIba. Despite its poor inhibitory activity against hepatic neuraminidase, there were several anecdotal reports demonstrating its success in treating ITP [91,92,93]. These preliminary findings led to the design of a prospective phase-2 study, the recently published results of which were relevant and encouraging. Newly diagnosed ITP patients were administered 40 mg dexamethasone daily for four days, either alone or with 75 mg oseltamivir daily for 10 days. The overall response (platelet count > 30 × 10^9^/L and at least doubling the baseline platelet count) was 66% at day 14 in the dexamethasone arm and, interestingly, 86% in the arm of those who also received oseltamivir. After six months, the overall response remained in 30 and 53% of patients in the dexamethasone and the oseltamivir plus dexamethasone groups, respectively. The difference between groups was noted at 12 and at 18 months. The authors stated that multiple cycles of oseltamivir might be more effective in maintaining the platelet response [94]. A phase-3 trial is currently being conducted with ITP patients, using a dosage pattern of 75 mg twice daily for five consecutive days. A positive result could pave the way toward a tailored ITP treatment that could substitute non-specific immune-blunting therapies, thus avoiding the lifelong summative immunosuppressive therapy burden.

### 3.8. Anti-CD38 Antibody

Plasma cells constitute the last step of B lymphocyte maturation [95]. B cells are responsible for the primary mechanism involved in ITP-triggered bleeding, i.e., platelet destruction via antibody production. CD38 is a cell surface molecule that is highly expressed on the surface of antibody-producing plasmablasts, on short-lived and long-lived plasma cells, natural killer (NK) cells, and also antigen-induced activated T and B cells [96]. In all these lineages, CD38 is involved in the key cell adhesion and signal transduction pathways. Therefore, drugs with the ability to block the CD38 protein and, subsequently, reduce cell viability may lead to the depletion of plasma cells and plasmablasts, along with a cytokine production decrease and, in the end, the subsequent decrease of immunoglobulin levels and autoantibody production. It is reasonable to think that monoclonal antibodies targeting the CD38 protein will be able to diminish CD38+ cells, thereby reducing the autoantibody-mediated destruction of platelets and, ultimately, improving platelet count (Appendix A) [97,98].

#### Daratumumab

Daratumumab is a subcutaneously administered, anti-CD38 monoclonal antibody already approved for multiple myeloma, which is able to induce apoptosis in CD38 highly expressing cells (Table 6 and Table 7). The preliminary results of an ongoing trial, the so-called DART Study, involving refractory ITP patients, are encouraging since they suggest that daratumumab can eliminate the autoreactive plasma cells involved in relapsing ITP [99]. Furthermore, a French group has recently published some interesting findings obtained with this drug in refractory ITP, albeit in a limited number of cases: daratumumab was able to attain an acceptable response rate (two responses out of five cases), in spite of the high refractoriness of the cohort, with a median of six previous treatments [100]. These results highlight the need for larger, randomized trials to establish the real potential of this type of drug in the scenario of refractory ITP. Indeed, an additional advantage of daratumumab is its long plasma half-life, which allows prolonged inter-dose intervals, with a subsequent gain in the patient’s quality of life.

#### Mezagitamab

Finally, it must be mentioned that, although less preliminary data is available, mezagitamab (TAK-079), which is another anti-CD38 monoclonal antibody, is also being investigated in the refractory ITP setting, in a trial that is currently ongoing.

### 3.9. Proteasome Inhibitors

The proteasome is a large intracellular protein complex that is responsible for the degradation of the majority of intracellular proteins. Its activity is finely tuned, and the gain or loss of function may lead to prolonged survival or premature cell death [101]. In refractory ITP, corticosteroid-resistant long-lived plasma cells play a key role in treatment failure. It is hypothesized that proteasome inhibitors targeting plasma cells could induce the elimination of these cells, thus reducing the generation of antiplatelet autoantibodies and the subsequent platelet destruction.

#### Bortezomib

Bortezomib is a small molecule of intravenous administration that is a well-known proteasome inhibitor (Table 6 and Table 7) (Appendix A). Bortezomib has been widely used in the treatment of multiple myeloma. In mice, bortezomib improved thrombocytopenia by destroying long-lived plasma cells [102]. Positive experiences with bortezomib in a few cases of refractory thrombotic thrombocytopenic purpura (TTP) [103], autoimmune hemolytic anemia (AIHA) [104], and, interestingly, one case of ITP [105] led to the design of a phase-2 trial on which there is currently no information available. Recently, the successful use of bortezomib in a child with refractory ITP has been reported [106].

#### Other Proteasome Inhibitors

Clinical trials with other proteasome inhibitors were launched but were not completed. The combination of the proteasome inhibitor ixazomib and dexamethasone was designed to be administered to ITP patients in the trial NCT03965624. However, the trial was withdrawn at the sponsor’s request. Finally, KZR-616 was used in a trial designed to recruit ITP and AIHA patients. KZR-616 is a selective, non-cytotoxic immunoproteasome inhibitor. The immunoproteasome is a proteasome that is expressed predominantly in the effector immune cells. Its inhibition would reduce the activity of Th cells, increase the number of Treg cells, and block the formation of plasma cells and autoantibodies. It is hypothesized that, unlike bortezomib, KZR-616 would not induce thrombocytopenia [107]. Unfortunately, the trial was suspended due to activity slowdown as a result of the COVID-19 pandemic, a high screen fail-rate, and a lack of enrollment (NCT04039477).

### 3.10. Therapies Targeting T or B Cells

CD40 is a membrane protein that belongs to the TNF-receptor superfamily and is expressed mainly in APCs and B cells. Its interaction with the CD40 ligand (CD40 L or CD154), which is expressed by CD4+ T cells and platelets, plays a key role in the cellular and humoral adaptive immunity processes. The expression of CD40 and CD154, as well as the levels of the soluble form of the latter (sCD40L), which has a cytokine-like activity, increase in inflammatory and autoimmune conditions, thus exacerbating the cellular and humoral immune responses. It is conceivable to think that halting the interaction between CD40 and CD154 may reduce the activity of autoreactive B cells and T cells [108]. One of the first drugs tested in this scenario was an anti-CD154 monoclonal antibody called toralizumab (IDEC-131). Initially, the results were encouraging in a phase-1 trial, where the treatment achieved a 60% response rate in chronic refractory ITP patients, with neither thrombotic events nor other severe side effects reported over a period of 3 months from administration [109]. However, in an animal model, the rate of thromboembolic events in those receiving toralizumab was high, possibly due, at least partly, to increased platelet-dependent coagulant activity. Accordingly, in phase-2 clinical trials regarding the use of this drug in multiple sclerosis and Crohn’s disease, thromboembolism events were reported in 3 patients. These findings finally led to the drug’s discontinuation. Nevertheless, although the investigation is in the very preliminary stages, there are new antibodies addressing the interaction between CD40 and CD154 that are candidates for use in ITP, and that are currently in ongoing phase-1 or 1/2 trials: letolizumab against CD154, and BI 655064 against CD40.

It is also worth mentioning that interleukin-21 (IL-21) may be a target of interest. IL-21 plays a key role in B cell activation and the subsequent autoantibody production. The anti-IL-21 monoclonal antibody BOS161721 is currently being investigated in other autoimmune disorders and could constitute a future tool for ITP management [110]. Other options at midterm may be the induction of Treg cells by low-dose interleukin-2, or monoclonal antibodies targeting CD80, CD86 or CTLA4-Ig.

### 3.11. Other Future Options

Other strategies may be of interest in the midterm, provided that the initial results, obtained in animal models or preliminary studies with a limited number of patients, are confirmed in larger trials.

#### 3.11.1. VPAC1 Inhibitors

The vasoactive intestinal peptide receptor 1 (VPAC1) is a receptor expressed on the surface of megakaryocytes and platelets. It has two ligands: the pituitary adenylyl cyclase-activating peptide (PACAP) and the vasoactive intestinal peptide (VIP). The activation of VPAC1 by PACAP inhibits megakaryocytic growth and differentiation. In a murine model, VPAC1-neutralizing antibodies were shown to reduce the intensity of thrombocytopenia [111].

#### 3.11.2. Amifostine

Amifostine is a small molecule of intravenous administration, used as a cytoprotective agent to prevent chemotherapy and radiotherapy-associated toxicity. Amifostine stimulates the hematopoietic stem cells and has been used with success in patients with myelodysplastic syndrome. In the ITP scenario, the first results were sound. In 24 patients with refractory ITP, 4 to 5 400-mg weekly infusions achieved persistent remission for 2 months in all patients [112].

#### 3.11.3. Atorvastatin

Endothelial cells are part of the bone marrow’s microenvironment. Bone marrow endothelial cells are capable of sustaining hematopoiesis, including megakaryocyte production. Subjects with refractory ITP have presented with a quantitative and qualitative alteration in their cell migration and angiogenesis capacity at the bone marrow level, as well as with increased reactive oxygen species (ROS) production and heightened cell-death level [113]. The use of atorvastatin may alter the described microenvironment in such a manner that platelet formation could be promoted. Very preliminary findings in refractory ITP patients show response rates of up to 60% [113]. Trials are underway, with atorvastatin alone or in combination with other drugs, to corroborate these findings (NCT03692754, NCT03460808).

#### 3.11.4. Epigenetic Modulation

Two drugs may be potentially useful for ITP management in this context. Decitabine is a small molecule of intravenous administration, used to treat myelodysplastic syndrome, which functions as a hypomethylating agent with a dual mechanism of action. On the one hand, it induces the reactivation of silenced genes and differentiation at low doses; on the other hand, it induces cytotoxicity at high doses. Low-dose decitabine has been shown to rebalance T-cell homeostasis, decrease proinflammatory cytokines, and downregulate phosphorylated STAT3 in ITP patients. Low-dose decitabine might restore Treg cells by inhibiting STAT3 activation and increasing platelet production. One study recruited 40 ITP patients to receive three cycles of low-dose decitabine. The percentage of Treg cells increased notably, while the circulating CD4+ T cells remained unaltered. No harmful cytotoxic effects were reported [114].

As mentioned earlier, the increased macrophage phagocytosis of antibody-coated platelets, as well as the decreased numbers and/or impaired function of CD4 + CD25 + Foxp3+ Treg cells, contribute to the pathogenesis of ITP. In vitro and in vivo experiments found that a low-dose histone deacetylase inhibitor (HDACi), the so-called chidamide, alleviated thrombocytopenia and attenuated the macrophage phagocytosis of antibody-coated platelets in a murine ITP model. The mechanism may involve stimulation of the production of natural Foxp3+ Treg cells, the peripheral conversion of T cells into Treg cells, and the restoration of Treg cell suppression in vivo and in vitro. Low-dose chidamide could also regulate CTLA4 expression in peripheral blood mononuclear cells through the modulation of histone H3K27 acetylation. The low-dose HDACi-based treatment of ITP patients could be offset by blocking the CTLA4 effect. These preliminary observations suggest that low-dose chidamide might have potential as a novel ITP treatment [115].

#### 3.11.5. Low-Level Laser Light

Platelet production from megakaryocytes requires a significant amount of energy, which is provided by mitochondria to power endomitosis, microtubule formation, organelle formation and their migration into proplatelets [116,117]. Megakaryocyte apoptosis may be accelerated in ITP, resulting in the reduction of platelet production [118]. A low-level laser light (LLL) activates cytochrome C oxidase in mitochondria and can thereby increase mitochondrial function by increasing the mitochondrial membrane potential and ATP, reducing apoptosis. When mature wild-type murine bone marrow megakaryocytes were exposed to LLL for 1 h, an increase in their size, together with proplatelet formation and platelet production, was reported by the third day [119].

## 4. Conclusions

ITP is a highly heterogeneous condition in terms of the patients’ response to treatment, which could be explained, at least partly, by the wide range of pathological mechanisms associated with the disease. Increasing the portfolio of available therapeutic options is highly desirable at any stage of the disease. Current first- and second-line treatments, although achieving reasonable response rates, are generally unable to sustain remission in the long term. Thus, patients may be re-exposed to bleeding risk. On the other hand, immunosuppressive therapies, if prolonged, increase the risk of infection, which may have severe consequences in fragile patients. Our increasingly deep knowledge of the mechanisms underlying the immune imbalances that cause ITP is allowing the design of a wide range of new drugs. Many of these are targeted against specific molecules or pathways, thus minimizing the side effects when compared with currently used therapies. The results of the ongoing trials may well change the guideline recommendations in the near future.

## Figures and Tables

**Figure 1 pharmaceuticals-15-00779-f001:**
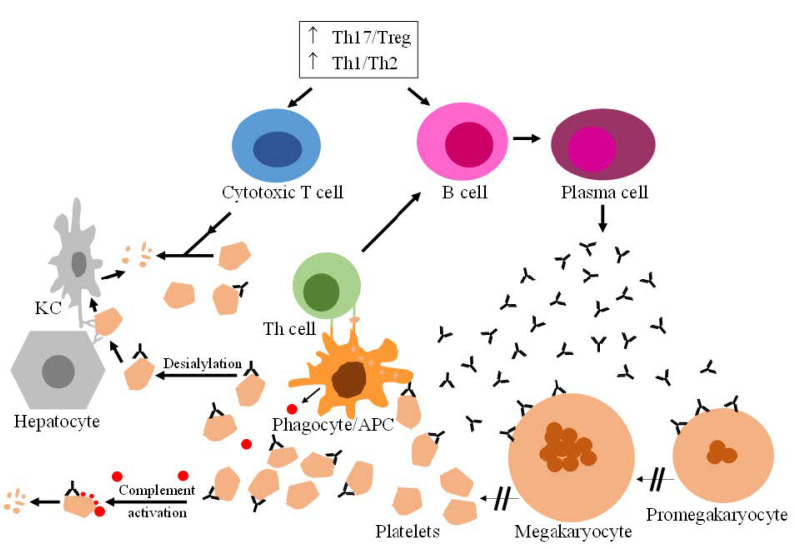
**Mechanisms of platelet destruction in ITP.** The immune imbalance leads to the excessive activation of cytotoxic T cells and autoreactive B cells. Plasma cell formation induces the release of antiplatelet autoantibodies. As a result, the opsonized platelets are destroyed by phagocytes, which can also present antigens to Th cells. Opsonized platelets can also be destroyed by the complement system; their surface glycoproteins are exposed to desialylation. This leads to subsequent platelet cleavage in the liver by Kupffer cells, after binding hepatocytes via Ashwell–Morell receptors. Finally, autoantibodies also interfere with platelet generation from megakaryocytes in the bone marrow. Note: APC, antigen presenting cell; KC, Kupffer cell.

**Figure 2 pharmaceuticals-15-00779-f002:**
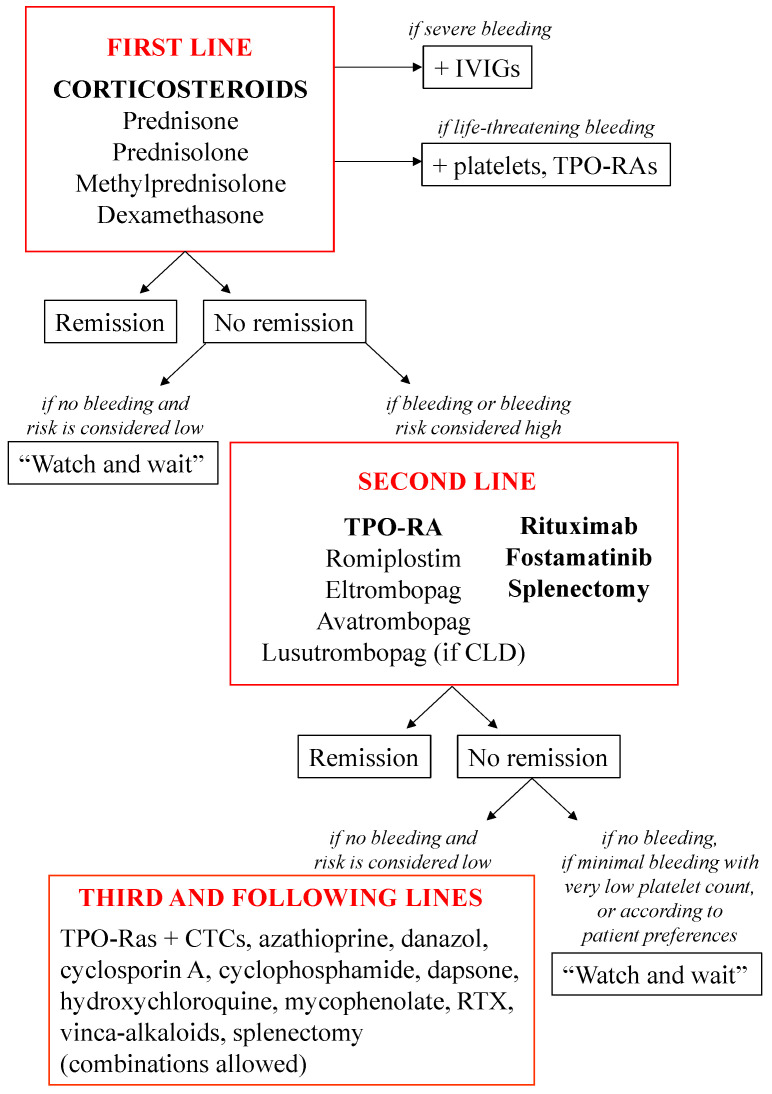
**Treatment preferences to manage ITP, according to the current guidelines.** Splenectomy should not be considered an option during the first 12 months after diagnosis. Third-line treatments are not necessarily arranged according to hierarchy. Note: CLD, chronic liver disease; CTC, corticosteroids; ITP, primary immune thrombocytopenia; IVIGs, intravenous immunoglobulins; RTX, rituximab; TPO-RAs, thrombopoietin receptor agonists.

**Table 1 pharmaceuticals-15-00779-t001:** Pharmacokinetics of the current first- and second-line drugs prescribed to treat ITP patients.

Drug	Absorption	Vd	PB	Elimination	t1/2	Clearance
**Prednisone**	Cmax: n.a. Tmax: 2 h	29.3 L (0.15 mg/kg dose); 44.2 L (0.30 mg/kg dose)	<50%	Urine	2–3 h	0.066 ± 0.12 L/h/kg (5.5 µg/h/kg dose)
**Prednisolone**	Cmax: 113–1343 ng/mL Tmax: 1.0–2.6 h Bioavailability: 70%	29.3 L (0.15 mg/kg dose); 44.2 L (0.30 mg/kg dose)	65–91%	Urine	2–3 h	0.09 L/kg/h (0.15 mg/kg dose) 0.12 L/kg/h (0.30 mg/kg dose)
**Methylprednisolone**	Bioavailability: 89.9%	1.38 L/kg	76.8%	Urine	2.3 h	336 mL/h/kg
**Dexamethasone**	Cmax: 13.9 ± 6.8 ng/mL Tmax: 2.0 ± 0.5 h (1.5 mg oral dose) Bioavailability: 70–78%	51.0 L (1.5 mg dose)	77%	<10% urine	6.6 h	15.6 ± 4.9 L/h (1.5 mg dose)
**Romiplostim**	Tmax: median 14 h (range: 7–50 h)	10 µg/kg (48.2 mL/kg)	n.a.	Renal (high doses), c-Mpl receptors (low doses)	median 3.5 d (range 1–34 d)	n.a.
**Eltrombopag**	Total absorption: ≥52% (75 mg oral dose) Tmax: 2–6 h	In blood cells, concentrations are 50–79% of those in plasma	>99%	Feces (59%), urine (31%)	26–35 h	n.a.
**Avatrombopag**	Cmax: 166 ng/mL (40 mg dose) Tmax: 5–8 h	180 L (CV 25%)	>96%	Feces (88%) urine (6%)	19 h	6.9 L/h (CV 29%)
**Rituximab**	Cmax: 157 ± 46 and 183 ± 55 ng/mL after 1st and 2nd infusion of a 500 mg dose (RA)	3.1 L (RA)	n.a.	Reticuloendothelial system	median 22 d (range 16–28 d)	0.335 L/d (RA)
**Fostamatinib**	Proportional exposure up to doses of 200 mg BD Tmax: 1.5 h Bioavailability: 55%	400 L	98.3%	Feces (80%), urine (20%)	15 h	300 mL/min

Part of the data has been obtained from the DRUGBANK online (https://go.drugbank.com/drugs, accessed on 23 April 2022). Note: data correspond to the active metabolite R406. BD, BD, twice a day; Cmax, peak plasma concentration; CV, coefficient of variation; d, days; h, hours; ITP, immune thrombocytopenia; n.a., not readily available; PB, protein binding; t1/2, plasma half-life; RA, in patients with rheumatoid arthritis; Tmax, time to reach Cmax; Vd, volume of distribution.

**Table 2 pharmaceuticals-15-00779-t002:** Pharmacodynamics of the current first- and second-line drugs indicated to treat ITP patients.

Drug	Pharmacodynamics
**Prednisone**	Inhibits pro-inflammatory signals and promotes anti-inflammatory signals by binding to the glucocorticoid receptor. Its duration of action is short, in agreement with its half-life of 2–3 h
**Prednisolone**	Inhibits pro-inflammatory signals and promotes anti-inflammatory signals by binding to the glucocorticoid receptor. Its duration of action is short, in agreement with its half-life of 2–3 h
**Methylprednisolone**	Inhibits pro-inflammatory signals and promotes anti-inflammatory signals by binding to the glucocorticoid receptor. Its duration of action is short, in agreement with its half-life of 2.3 h
**Dexamethasone**	Inhibits pro-inflammatory signals and promotes anti-inflammatory signals by binding to the glucocorticoid receptor. Its duration of action is somewhat longer than that of other glucocorticoids, in agreement with its longer half-life
**TPOa** **(Romiplostim, Eltrombopag, Avatrombopag)**	Drug dose-dependency has been reported for platelet increase. The extent of the effect may vary among patients, which means that dose individualization is required.
**Rituximab**	Binding to the CD20 antigen on mature B cell surfaces induces the selective killing of B cells. More details on pharmacodynamics are available for other autoimmune conditions. In RA, the near-complete depletion of peripheral B cells is achieved within 2 weeks after the first dose, which may be sustained for ≥6 months
**Fostamatinib**	The active metabolite R406 inhibits signal transduction by Fcγ receptors involved in the antibody-mediated destruction of platelets by immune cells, thus increasing platelet counts. R406 inhibits T and B lymphocyte activation by T-cell and B-cell receptors. The inhibition of Fc receptor signaling suppresses dendritic cell maturation and antigen presentation. Production of the major inflammatory mediators and cytokines is also reduced

Part of the information has been obtained from the DRUGBANK online (https://go.drugbank.com/drugs, accessed on 23 April 2022). Note: CLD, chronic liver disease; d, days; h, hours; TPOa, thrombopoietin analogs.

**Table 3 pharmaceuticals-15-00779-t003:** Dosage, response rate, and side effects of the current first- and second-line drugs indicated to treat ITP patients [8,10,11].

Drug	Dose	Onset of Action/ Peak Response (d)	Overall Response	Side Effects
**Prednisone or prednisolone**	0.5–2 mg/kg/d p.o. for 2–3 wk, gradually tapered next 6–8 wk. Rapid 2 wk tapering in case of no response	4–14/7–28	In the short term: 60–80%. Sustained after drug discontinuation: 30–50%.	Weight gain, cushingoid phenotype, infection, hypertension, hyperglycemia, hirsutism, cataracts, mood disorders
**Methylprednisolone**	1 g/d i.v. for 3 d (or 15 mg/kg/d)	2–14/4–28	Similar to those of prednisone or prednisolone	Similar to those of prednisone or prednisolone
**Dexamethasone**	40 mg p.o. or i.v. from D1 to D4, up to 3–4 cycles, each cycle after 2–4 wk	2–14/4–28	More rapid response than prednisone, but similar in the long term	Weight gain and cushingoid phenotype more attenuated, and infection rate lower than that observed with prednisone
**IVIG ^a^**	1 g/kg/d i.v. for 1–2 d or 400 mg/kg/d i.v. for 5 d	1–3/2–7	n.a.	Headache, anaphylaxis
**Romiplostim**	1–10 μg/kg/wk s.c., initially with the minimum dose. Titration must be according to platelet response	7–14/16–60	70–80% (maintenance therapy required)Sustained after discontinuation: 10–30%	Pain at injection site, body ache, headache, thrombosis, bone marrow fibrosis, reticulin increase
**Eltrombopag**	25–75 mg/d p.o. (2 h before or 4 h after food or polyanion (Ca, Fe)-containing products)	7–14/16–90	Similar to romiplostim	Transaminitis, gastrointestinal discomfort, thrombosis, bone-marrow fibrosis
**Avatrombopag**	5–40 mg/d p.o.	3–5/10–13	65% on D8 of treatment	Thrombosis (rarely), arthralgia, headache
**Rituximab**	375 or 100 mg/m^2^/wk i.v. for 4 wk	7–56/14–180	At short term: 60–80%. At long term (2–5 yr): 20–30%	Infusion related reactions: fever, chills, rigor. At long-term: hypogammaglobulinemia, infection, reactivation of hepatitis B, PML
**Fostamatinib**	50–150 mg p.o. BD	7–14/n.a.	18–43%	Diarrhea, hypertension, infection

Note: ^a^ as a rescue treatment, in the case of severe bleeding. BD, twice a day; D, day; d, days; i.v., intravenous; n.a., not available; PML, progressive multifocal leukoencephalopathy; p.o., oral; s.c., subcutaneous; wk, weeks; yr, years.

**Table 4 pharmaceuticals-15-00779-t004:** New therapies that may contribute to filling unmet needs in ITP treatment.

Gaps to Fill Regarding ITP Treatment
Increase in the efficacy of new drugs, redirecting research toward targets where modulation results in a more durable improvement
Increased sustained response to first-line treatment
New immune system modulation-based treatments may shed light on the mechanisms underlying distorted immune tolerance, thus allowing the use (or future design) of more specific drugs
Analyzing the efficacy of new drugs targeting specific mechanisms should shed light on their relevance, unveiling those pathways on which therapies should focus

Note: ITP, primary immune thrombocytopenia; TPO-RAs, thrombopoietin receptor agonists.

**Table 5 pharmaceuticals-15-00779-t005:** ITP treatments that are under clinical investigation.

Type of Approach	Mechanism of Action	Drug-Development Stage (Finished/Ongoing Trials)
**TPO-RAs**	Increase in megakaryocytes and subsequent platelet production	Hetrombopag - Phase 1, NCT02403440 - Phase 3 (children), NCT04737850
**Syk inhibition**	Inhibition of macrophage phagocytosis and the subsequent decrease in platelet destruction	HMPL-523 - Phase 1B, NCT03951623 - Phase 3, NCT05029635 SIK-O-703, - Phase 1, NCT02717988 - Phase 1, NCT03315494 - Phase 2, NCT04056195
**BTK inhibition**	Inhibition of macrophage phagocytosis and the subsequent decrease in platelet destruction	Rilzabrutinib - Phase 1, NCT03395210 - Phase 3, NCT04562766 Orelabrutinib - Phase 2, NCT05232149 - Phase 1, 2 (refractory) NCT05020288
**FcRn blockers**	Increase in antiplatelet autoantibody clearance, thus decreasing the peripheral platelet destruction and immune response against megakaryocytes	Efgartigimod - Phase 3, NCT04225156 - Phase 3 (sc), NCT04687072 - Phase 3 (sc), NCT04812925 Rozanolixizumab - Phase 3, NCT00718692 - Phase 3, NCT04200456 HBM9161 (Batoclimab) - Phase 2/3, NCT04428255
**Staphylococcal protein A**	Inhibition of macrophage phagocytosis by preventing FcγRIII (CD16) participation	PRTX-100 - Phase 1/2, NCT02401061 - Phase 1/2, NCT02566603
**Immunoglobulin-based drugs**	Decrease in platelet destruction by splenic macrophages and increase in antiplatelet autoantibody clearance by FcRn saturation	M254 - Phase 1, NCT03866577 PF06755347 - Phase 1, NCT03275740 M230 - Phase 1, NCT04446000
**Complement inhibition**	Decrease in complement-dependent cytotoxicity	Sutimlimab - Phase 1, NCT03275454
**Neuraminidase inhibition**	Inhibition of platelet desialylation prevents liver platelet destruction	Oseltamivir - Phase 2, NCT01965626 - Phase 3, NCT03520049
**Anti-CD38 inhibition**	Reduction of autoantibody production and immune imbalance by the inhibition of CD38 on the surface of plasma cells and other immune cells	Daratumumab - Phase 2, NCT04703621 Mezagitamab - Phase 2, NCT04278924
**Proteasome inhibition**	Decreased autoantibody production by preventing long-lived plasma cells	Bortezomib - Phase 1 (NCT03013114)
**Interference with the interaction between CD40 and CD154**	Partial resolution of the imbalance between cellular and humoral adaptive immunity processes	Letolizumab - Phase 1/2, NCT02273960 BI 655064 - Phase 1, NCT02009761

Note: BTK, Bruton’s tyrosine kinase; Syk, spleen tyrosine kinase; TPO-RAs, thrombopoietin receptor agonists.

**Table 6 pharmaceuticals-15-00779-t006:** Pharmacokinetics of drugs under clinical investigation to treat ITP patients.

Drug	Absorption	Vd	PB	Elimination	t1/2	Clearance
**Hetrombopag**	Cmax: 24 ng/mL (5 mg oral dose) Tmax: 8 h	n.a.	n.a.	Feces (62.5)	11.9–40 h	15.6 L/h
**HMPL-523**	Cmax: dose-dependent until 800 mg dose Tmax: 3–6 h	n.a.	n.a.	n.a.	9.8–13.5 h (dose range tested: 100–800 mg)	n.a.
**Rilzabrutinib**	Cmax: 91 ng/mL (300 mg dose) Tmax: 1.5 h	4910 L (300 mg dose)	n.a.	n.a.	1.3–3.9 h (dose range tested: 50–1200 mg)	1.6%/h (occupancydecay rate)
**Efgartigimod**	Exposure increases proportionally up to 50 mg/kg	15–20 L	n.a.	Proteolytic enzymes (urine < 0.1%)	80–120 h	n.a.
**Rozanolixizumab**	(i.v.) Cmax: 89–154 µg/mL (4–7 mg/kg dose) Tmax: 1–2.5 h (s.c.) Cmax: 12 µg/mL (7 mg/kg dose) Tmax: 48 h	n.a.	n.a.	Predominantly by reticuloendothelial macrophages	n.a.	n.a.
**Sutimlimab**	Exposure increases proportionately with increasing dosage. Steady-state achieved by 7th week	5.8 L	n.a.	Predominantly by reticuloendothelial macrophages	21 d (6.5–7.5 g i.v. dose)	0.14 L/d
**Oseltamivir**	Cmax: 65 ng/mL (oral 75 mg BD)	23–26 L	42%	Renal excretion (>90%), feces (<20%)	1–3 h (oral 75 mg BD)	18.8 L/h
**Daratumumab**	Cmax: 592 µg/mL (1800 mg s.c. dose)	5.2 L (central compartment), 3.8 L (peripheral compartment)	n.a.	Predominantly by reticuloendothelial macrophages	20 d	119 mL/d
**Bortezomib**	Cmax: 57–112 ng/mL (1–1.3 mg/m^2^ i.v. dose)	498–1884 L/m^2^ (1–1.3 mg/m^2^ i.v. dose)	83%	Renal and hepatic routes	40–193 h (1 mg/m^2^ i.v. dose)	102–112 L/h (1–1.3 mg/m^2^ i.v. dose)

Part of the data has been obtained from the DRUGBANK online (https://go.drugbank.com/drugs, accessed on 23 April 2022). Note: BD, twice a day; Cmax, peak plasma concentration; d, days; h, hours; ITP, immune thrombocytopenia; i.v., intravenous; n.a., not readily available; PB, protein binding; s.c., subcutaneous; t1/2, plasma half-life; Tmax, time to reach Cmax; Vd, volume of distribution.

**Table 7 pharmaceuticals-15-00779-t007:** Pharmacodynamics of the drugs under clinical investigation to treat ITP patients.

Drug	Pharmacodynamics
**Hetrombopag**	Dynamic changes in blood platelets are best characterized by four-transit compartment models.Single dose (5–40 mg): a consistent increase in platelet counts is observed after D4, the maximal effect being reported on D10. The thrombopoietic activity is dose-dependent, with platelet counts increasing twofold with a dose of 40 mg.Repeated daily doses (2.5–7.5 mg) for 10 d: an increase in platelet counts starts after 6 d, with the peak at 12–14 d. Eighteen days after the last dose, platelet counts are still 18.8% and 32.2% above baseline in those patients administered 5 and 7.5 mg doses, respectively
**HMPL-523**	Inhibits anti-IgE-induced basophil (CD63+) dose-dependently, with an EC50 of 47.70 ng/mL
**Rilzabrutinib**	Occupancy of BTK occurs rapidly and dose-dependently, with doses ≥ 150 mg and the maximum occupancy (>90%) within the first 4 h. A 30–35% reduction in occupancy is observed with all doses between hours 4 and 24. IC50 = 1.3 nM
**Orelabrutinib**	Near-complete BTK occupancy is achieved at doses ≥ 50 mg (Cmax to achieve EC99: 300 ng/mL), the effect being sustained for 24 h post-dosing, which is consistent with the covalent binding mode of action. The IC50 value is 1.6 nM
**Efgartigimod**	The pharmacologic effect is exerted by reducing the circulating levels of autoantibodies. Because efgartigimod also reduces the level of the rest of the IgGs, patients may be at greater risk of infection. Treatment should not be initiated in patients with an active infection. Accordingly, efgartigimod discontinuation should be considered in patients who develop a serious infection during therapy
**Rozanolixizumab**	A reduction in total serum IgG concentration over time is observed with both i.v. and s.c. administration, in a dose-dependent manner. Similar maximum reductions are observed via i.v. or s.c. The greatest IgG reduction is reported to be seen by days 7 to 10, the baseline level being restored by day 57. Reductions in the serum IgG on day 10 are 14.5, 33.4, and 47.6%, with 1, 4, and 7 mg/kg doses in the case of i.v. administration, and 16.8, 25.9, and 43.4% when s.c. administration is used
**Sutimlimab**	After a single i.v. injection, > 90% inhibition of the complement pathway is observed, which is sustained for concentrations of sutimlimab ≥100 µg/mL. The impairment in complement-mediated immune response makes necessary appropriate vaccination against encapsulated bacteria at least 2 weeks prior to treatment initiation. Since patients are at a higher risk of serious infections, they have to be closely monitored throughout therapy
**Oseltamivir**	Once hydrolyzed to its active metabolite oseltamivir carboxylate, the drug exerts neuraminidase inhibitor activity, via competitive inhibition of the activity of neuraminidase upon sialic acid, which is found on glycoproteins on the surface of platelets. By blocking the activity of the enzyme, platelet destruction in the liver may be prevented
**Daratumumab**	Apoptosis is induced in CD38 highly expressing cells. The long duration of action allows dosing on a weekly basis. It is advisable to counsel patients regarding the risk of neutropenia, thrombocytopenia, embryo-fetal toxicity, hypersensitivity, and interference with cross-matching and red blood cell antibody screening
**Bortezomib**	The target is the ubiquitin-proteasome pathway, which regulates intracellular concentrations of proteins and promotes protein degradation, and may be dysregulated in pathological conditions. By reversibly inhibiting proteasome, proteasome-mediated proteolysis is prevented. Inhibition occurs in a dose-dependent manner

Part of the information has been obtained from the DRUGBANK online (https://go.drugbank.com/drugs, accessed on 23 April 2022). Note: D, day; d, days; EC50, estimated half-maximal effective concentration; EC99, concentration required to achieve >99% occupancy of the target; h, hours; IC50, half-maximal inhibitory concentration; i.v., intravenous; s.c., subcutaneous.

## Data Availability

Not applicable.

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
