# Peer review of "Novel Therapies to Address Unmet Needs in ITP"

_pharmaceuticals, 2022, doi:10.3390/ph15070779_

Round 1
Reviewer 1 Report
There is a number of similar published review on the management of ITP in adults including novel therapies (Audia and Bonnotte 2021, Singh et al 2021, Kuter 2022, Al-Samhari & Kuter 2020....) with the international consensus report published by Provan et al. in 2019. Although a big work done, this review is not a significant contribution to the fiels.
It is stated in Abstract that extensive pathophysiology mechanisms are described, but they are only mentioned.

Author Response
Dear colleague, I would like to explain you that it has been the magazine itself that has requested this document from us. I understand the interest in addressing this issue from its editorial line. We were requested for the delivery date in May 2022, and we did it. After sending the document for the first time, we were recommended to review the mechanism of action in deeper way, and we have done so, delivering the document one month from the initial shipment as indicated. In the review, the content has been expanded by almost a third, reviewing the depth of what has been indicated to us and sending it back to evaluation. I believe that the comment you send us should be aligned with the editor and the journal opinion, since if it was not of interest, perhaps the effort made should not be requested. In any case, the work has been carried out with great care and the desire to help colleagues who are interested in the subject with the vision of the Spanish national PTI working group combining clinical and pharmaceutical point of view which is value of this individual paper.

Reviewer 2 Report
This is a comprehensive and excellently written manuscript on novel therapies of patients with ITP including a review of currently used drugs. There are numerous figures and tables in the manuscript and as supplemental materials with clearly arranged presentations of immunological pathways and drugs interacting with them. There are no major points to criticize.
In Tables with two columns, such as table 2 describing pharmacodynamics, it is recommend to harmonize the contents, as there is sometimes quite different information, e.g. the description of the three TPO-RAs.
In recommending the time point of splenectomy, it is stated in the legend for figure 2, to do that not earlier than 12 months and in 2.2.4. (splenectomy) it is stated that it should be avoided in the first 24 months, please be consistent in this important statement.
For predictability of a successful splenectomy, it is recommend to at least mention platelet scintigraphy, as this method has been developed successfully with good prognostic value, but with low availability.
Page 2, first paragraph, last line: …maturation…
Author Response
Reviewer 2 answers:
Dear collague, thanks for all your helpful observations. Here you are the modifications.
In Tables with two columns, such as table 2 describing pharmacodynamics, it is recommend to harmonize the contents, as there is sometimes quite different information, e.g. the description of the three TPO-RAs.
We have include in the same group description the three TPOa.
In recommending the time point of splenectomy, it is stated in the legend for figure 2, to do that not earlier than 12 months and in 2.2.4. (splenectomy) it is stated that it should be avoided in the first 24 months, please be consistent in this important statement.
We have change to 12 months in the test (point 2.2.4)
For predictability of a successful splenectomy, it is recommend to at least mention platelet scintigraphy, as this method has been developed successfully with good prognostic value, but with low availability.
We have introduce the comment in the document. “However, the identification of those ITP patients who would benefit most from this procedure has been precluded by the lack of reliable predictors of response. Platelet scintigraphy studies, have been describe for some groups a useful tool with high response rate in patients with close to pure splenic profile. Despite of this, results are controversial and there is low availability, so in guideline is not recommended [49]”.
Page 2, first paragraph, last line: …maturation…
We have change the word.

Reviewer 3 Report
Authors reviewed the novel therapies to address unmet needs in ITP.
This manuscript is potentially interesting, several issues arise.
1. Authors should state the sterilization for helicobacter pylori in patients with ITP.
2. It is helpful to show the platelet counts in patients with ITP who should be treated for ITP.
3. Figure 1: Is liver more important than spleen?
4. Figure 2: Platelet counts may be helpful when the treatment starts.
5. Table 2: Many abbreviations should be explained.
6. Table 3: References may be helpful.
7. Table 4 is not clear.
Author Response
REVIEWER 3
Dear collague, first of all, thanks for all your helpful observations. Here you are the modifications and comments.
- Authors should state the sterilization for helicobacter pylori in patients with ITP.
We have included a comment about HP.
“In Helicobater pylori infection should be evaluated in appropriate geographical areas and treated in case of positive test because of the possibility of ITP secondary to this infection and the low incidence of adverse events related with this recommendation [10]”
- It is helpful to show the platelet counts in patients with ITP who should be treated for ITP.
Thanks for the comment, but the objective of this document is to review new drugs in ITP not exactly a document of recommendations. If we include the number of platelets, we will have mention other topics related with indication of treatment like bleeding, patients characteristics, anticoagulation, antiplatelet treatment and so. Any way, if you think this is necessary, please confirmed it and we will do it.
- Figure 1: Is liver more important than spleen?
In the figure we do not describe the hierarchy or the importance of one pathophysiological pathway or another in ITP, above all because the weight can be different depending on the patient.
- Figure 2: Platelet counts may be helpful when the treatment starts.
Thanks for the comment, but as we mention before, the objective of this document is to review new drugs in ITP not exactly a document of recommendations. If we include the number of platelets, we will have mention other topics related with indication of treatment like bleeding, patients characteristics, anticoagulation, antiplatelet treatment and so. Any way, if you think this is necessary, please confirmed it and we will do it.
- Table 2: Many abbreviations should be explained.
Dear reviewer, we have gone through the Table 2, abbreviations have been explained.
- Table 3: References may be helpful.
Dear reviewer, These are the standard doses described in guidelines and recommendations, and it is dense to include all the papers, so we have included reference to guidelines to give an answer to your observation.
- Table 4 is not clear.
We have change Table 4 with the gaps to solve with the new ITP drugs/treatment options.
